# Brain-Computer Interface-Based Humanoid Control: A Review

**DOI:** 10.3390/s20133620

**Published:** 2020-06-27

**Authors:** Vinay Chamola, Ankur Vineet, Anand Nayyar, Eklas Hossain

**Affiliations:** 1Department of Electrical and Electronics, Birla Institute of Technology & Science, Pilani 333031, India; vinay.chamola@pilani.bits-pilani.ac.in (V.C.); h20180144@pilani.bits-pilani.ac.in (A.V.); 2Graduate School, Duy Tan University, Da Nang 550000, Vietnam; anandnayyar@duytan.edu.vn; 3Faculty of Information Technology, Duy Tan University, Da Nang 550000, Vietnam; 4Department of Electrical Engineering and Renewable energy, Oregon Institute of Technology, Klamath Falls, OR 97601, USA

**Keywords:** brain-computer interface (BCI), data fusion, nao humanoid, electroencephalography (EEG), P300, biological feedback

## Abstract

A Brain-Computer Interface (BCI) acts as a communication mechanism using brain signals to control external devices. The generation of such signals is sometimes independent of the nervous system, such as in Passive BCI. This is majorly beneficial for those who have severe motor disabilities. Traditional BCI systems have been dependent only on brain signals recorded using Electroencephalography (EEG) and have used a rule-based translation algorithm to generate control commands. However, the recent use of multi-sensor data fusion and machine learning-based translation algorithms has improved the accuracy of such systems. This paper discusses various BCI applications such as tele-presence, grasping of objects, navigation, etc. that use multi-sensor fusion and machine learning to control a humanoid robot to perform a desired task. The paper also includes a review of the methods and system design used in the discussed applications.

## 1. Introduction

Brain-Computer Interfaces (BCIs) lie at the intersection of signal processing, machine learning, and robotics systems. Brain-Computer Interface is a technique that records and processes the brain signals of a person to perform a desired actuation. Electroencephalography (EEG), Electrocorticography (ECoG), and Near-Infrared Spectroscopy (NIRS) are a few methods used for the recording brain signals. However, EEG is one of the most common methods used for BCI applications [1,2]. BCI provides an opportunity to develop a new form of communication mechanism controlled using brain signals. This kind of mechanism becomes extremely helpful for those with motor impairment [3]. For example, applications such as brain-controlled limbs, brain-controlled chairs, brain-controlled speech systems, etc. can be developed using a Brain-Computer Interface.

Combining this communication mechanism and interfacing with a humanoid robot opens up several possibilities to replicate human actions. A humanoid robot [4,5] resembles the human body in terms of the shape and range of actions it can perform. This makes the humanoid robot a perfect candidate for receiving the actuation from the brain signals and then interacting with its environment accordingly. Since a humanoid robot is almost a replica of a human being, it can be controlled to perform various day-to-day tasks that a human being performs. Thus humanoids have great potential with a large number of prospective day-to-day applications of which they can perform. Such humanoids can especially serve as assistants for the disabled by helping them with their daily activities. Humanoid systems can also be used in mission-critical operations like disaster recovery [6,7], military operations [8,9,10], etc. However, the reliability of the system required in such applications is much more than that of previous applications. Security of such systems is also a major concern. Hence, there has been growing research in this direction to secure such systems, thereby avoiding their being hacked and misused [11,12,13].

While designing a BCI-controlled humanoid, the brain-control interface system requires a translation algorithm to convert the input brain signals to generate control signals for the humanoid. Traditionally, brain signals were solely taken as an input signal for this purpose. However, at times it suffered from long training time and poor accuracy. One of the major factors that contributed to this was the significant variation in the input signal. To improve the performance of such systems, researchers have actively explored multi-sensor fusion in the past several years. Such systems are often termed as hybrid BCI systems and they make control decisions based on the fusion of inputs from various sensors. The use of this multi-sensor fusion has been shown to improve the robustness of the BCI-based system [14,15]. The major contributions of this paper are as follows:This paper reviews various applications in which a humanoid is controlled using brain signals for performing a wide variety of applications such as grasping of objects, navigation, telepresence etc.;For each of the applications, we discuss the overview of the application, system design, and results associated with the experiments conducted;Specifically in this review, we consider BCI applications which use just EEG signals (discussed in Section 3), applications which use multisensor fusion where in addition to EEG, other sensor inputs are also considered for execution of the desired task (Section 4), as well as augmented reality-assisted BCI (Section 5);To the best of our knowledge, this work is the first review on BCI-controlled humanoids.

The rest of the paper is organised as follows: Section 2 discusses the preliminary knowledge required to understand the paper. Section 3 discusses applications where a humanoid robot is controlled using only brain signals. Section 4 discusses humanoid control applications using hybrid BCI. Section 5 discusses a BCI-controlled humanoid application supported by Augmented Reality. Section 6 summarises the applications discussed in the paper. Section 7 concludes the paper.

## 2. Preliminary Knowledge

This section discusses a few preliminary basics that are required to understand the works described in the paper.

### 2.1. Brain-Computer Interface

Rehabilitation is one of the major areas where BCI finds its applications. BCI can act as a communication mechanism for those with motor impairment. In the case of people with motor impairment, their nervous system is not able to execute as per the brain’s signals. For example, the brain may think of lifting the left hand, but due to a person’s left hand being paralyzed (on account of nervous disorders), the hand may be unable to move. However, the signals from the brain can be directly sensed using EEG electrodes, and can be used to control a robotic arm which may imitate the lifting of the left arm [16]. Various works like [17,18,19,20,21,22,23,24,25,26,27,28,29,30] discuss several BCI applications. Such applications have greatly motivated recent advances in BCI, for it offers new communication possibilities for those who are paralyzed or suffer from various bodily disabilities. BCI works in three stages. The first stage involves taking input from the brain, which is generally done using Electroencephalography (EEG). The second stage consists of a translation algorithm that maps the input signals from the brain to a predefined output command, and the third stage involves controlling the external device based on the command [31,32,33].

Next we discuss these three stages in a little more detail. BCI Input (the first stage): This stage consists of acquiring data pertaining to one or more features of the brain’s activity. Different parts of the brain are responsible for processing different functions. For example, sensory functions related to vision are processed in the occipital lobe of the brain. Furthermore, the frontal lobe is responsible for planning, decisions, and making speech [34]. Depending on the desired action to be performed, the EEG sensors can be used to acquire the brain signals from that portion of the brain for further processing. The second stage namely the Translation Algorithm, takes the acquired brain signals as the input, and translates them into a specific output command, which could be used for a particular action. In particular, this stage involves using various classification algorithms like Linear Discriminant analysis (LDA), Artificial Neural Networks (ANN), etc. (as discussed in Section 2.3) for classifying the action into a particular category. The key features of the translation algorithm are the transfer function used, its adaptability, and the control output generated. The transfer function can be linear (e.g., LDA) or non-linear (e.g., Neural-Network). Adaptive algorithms can use sophisticated machine-learning algorithms to adapt according to the brain [35]. The third stage, BCI Output, deals with the output. The control output generated for application-specific devices can be of two forms: (i) Discrete or (ii) continuous. The discrete output is the one that can be used for selection among fixed outputs (e.g., letter selection) while the continuous output can help in navigating (e.g., cursor movement) [17].

### 2.2. Hybrid BCI

Traditional BCI approaches were dependent on just using brain signals for generating output. However, it is observed that the salient features of the brain signals could differ among various subjects. In fact, sometimes even for the same subject, the features varied from trial to trial [36]. Also analyzing a single aspect or feature can, at times, lead to missing out important information. These challenges make the use of machine learning for specifying and extracting features from the signals very appropriate. Machine learning has been used in various areas of application in the past to solve challenges of diverse natures [37,38,39] and also find great applicability in solving challenges related to BCI signals. Machine learning methods have been able to increase the decoding accuracy prominently as discussed later in the paper. To maximise the robustness of the system, to increase the information transfer rate, and to decrease the training time, th BCI system records and analyzes multiple complementary signals [40,41]. These systems use data fusion techniques and use machine-learning algorithms for the fusion of complementary signals. This technique is termed a Hybrid BCI, as demonstrated in Figure 1.

Any Hybrid BCI system must fulfil four major criteria, that are as follows [42,43]:Brain signals must be used in the BCI System;The user should be able to control one of the brain signals intentionally;The BCI System should do real-time processing of the signal;User must be provided with the feedback of the BCI output.

Generally combinations of a signal used by Hybrid BCI include a mix of Electromyography [44] (EMG) + Electroencephalography (EEG), Event-Related Desynchronization (ERD) along with Steady State Visual Evoked Potential (SSVEP), Near-Infrared Spectroscopy (NIRS) along with EEG, ERD along with P300 etc [45,46]. Table 1 lists the description of the major signals and methods discussed above [47,48,49].

### 2.3. Classification Algorithms

A major requirement of the classifiers in the BCI systems is to ensure good performance in terms of classification accuracy [50]. For example, let us take the case of a patient using a BCI-controlled wheelchair. Now suppose they have the facility to control the BCI wheelchair by taking it left, right, front, or back based on their thoughts. So when they think that the wheelchair should move left, the BCI system should be able to process the brain signals appropriately and must classify the action to be ‘move left’. This classification algorithms have the task of taking multiple features (e.g., brain signals) as an input and to distinguish between different classes (e.g., left, right, front, back in the example given here). In performing this task, it is important to choose features carefully so that the classification algorithm can significantly differentiate between the multiple classes [51]. The feature that acts as an input to the BCI system for controlling humanoid robots are of two types: (i) Temporal features or (ii) frequency features. Temporal features represents the amplitude of the generated signals with time, whereas frequency features represent the frequency power spectra of the signals. Generally, P300-based BCI uses temporal features whereas ERD- and SSVEP-based BCI uses frequency features.

*Classification*: Different classifiers are used to translate the features extracted from brain signals to control commands [52,53,54,55]. These classifiers range from the simplistic linear classifiers to complex non-linear classifiers. Some of the commonly used classifiers are: (i) Linear Discriminant analysis (LDA), (ii) Support Vector Machines (SVM), (iii) Artificial Neural Networks (ANN), and (iv) Statistical classifiers [56]. These classifiers are discussed in detail below.

**Linear Discriminant analysis (LDA) [57]:** LDA is a type of linear classifier. The major benefits of using LDA is that: (i) The computational complexity of LDA is less, and hence the time taken for the classification is reduced. This is useful when using the algorithm in an online session as discussed later. (ii) LDA is a simple classifier to use and visualise. Linearity can be a limitation while handling non-linear EEG data. On the other hand simpler techniques like LDA are suitable when small training data set is available. LDA is used in a number of BCI-controlled humanoid applications for classification. Typical decision boundary of LDA is shown in Figure 2. For LDA, decision boundary are singly connected and convex. Figure 2 denotes 3 class classification in which the colour of the region denotes the class being predicted.

**Artificial Neural Networks (ANN) [58,59]:** ANN is a type of non-linear classifier. The classifier is inspired by the neuron structure of the brain. It is used to approximate non-linear functions. Using ANN is generally computationally intensive and requires a number of parameters to be configured. It is more complex in terms of usage as compared to LDA and the computational time taken to generate the output is also longer. However, ANNs are highly adaptive and can be applied on a wide variety of use-cases. Unfortunately, ANNs are prone to over-fitting, and thus the selection of the parameters/architecture and regularisation needs to be done carefully. The decision boundary of ANN can be seen in Figure 2, the non-linearity of the function is evident from the figure. The figure shows two classes, one represented using red colour and the other one using a blue colour that has been classified using ANN.

**Support Vector Machines (SVM) [57,60]:** SVM is also a non-linear classifier. However, while using SVM, setting up of the configurations is not needed. It is useful in cases when the training data is less. Most of the time it generalises better. This makes its use advantageous for BCI systems as the classifiers once trained, classify brain signals for multiple sessions. The features generated during multiple sessions may vary even for a single user. Hence the models which are less sensitive to over-fitting may perform better. SVM also performs well with high dimensionality data. However, SVM are sometimes slower than other classifiers, which becomes an issue while dealing with large data. Decision boundary with maximising margin between the classes is shown in Figure 2.

**Statistical Classifiers:** These classifiers [61] use posterior probabilities to select the class that has the highest probability based on the input features of every new instance. This type of classifiers utilise prior knowledge to classify instances. These classifiers also perform well in case of uncertainty, which is expected when dealing with brain signals. Uncertainty of the signals can be caused by fatigue or learning effects.

Table 2 summarises the typical classifiers that are applied in BCI.

### 2.4. Humanoids

A humanoid robot is a robot with a body structure and features similar to that of a human. Three main primitives for a humanoid robot are sensors, planning, and control. Humanoid robots generally have proprioceptive sensors to sense the position and exteroceptive sensors to get data on what is being touched. Actuators in humanoid robots mimic the action of muscles and joints. Following is a list of the humanoid robots, which have been commonly used in BCI-controlled humanoid applications in the recent past as shown in Figure 3. NAO (Nao Humanoid) humanoid [62], which is developed by Softbank robotics is one of the most commonly used and is actively used for research and educational purposes.

Nao Humanoid (Softbank Robotics) [62];HRP-2 Humanoid (Kawada Industries) [63];KT-X Humanoid (Kumotek Robotics) [24];DARwIn-OP (Robotis) [64].

In general, the humanoid robots in the list above have the following set of characteristics:17–30 degrees of freedom;Multiple sensors like gyroscope, force sensors, etc. on different body parts like head, torso, arms, legs;Microphones and speakers to interact with humans;Two cameras for object detection and recognition (in NAO);Custom application development due to open architecture.

Figure 4 gives an overview of the BCI-controlled humanoid applications discussed in the paper. Majorly, P300 signal is used in these applications as it gives high accuracy [48,65].

## 3. BCI-Controlled Humanoid Applications Using Only EEG

In this section, we discuss various BCI-controlled humanoid applications that use only the EEG signal as an input. The EEG input is processed and translated to an appropriate control output. Specifically, we consider three applications, namely grasping a glass of water, telepresence, and museum guide application using the BCI-controlled humanoid. These applications are discussed in the following subsections one by one. For every application, we provide an overview, system design description followed by the salient results associated with the conducted experimentation.

### 3.1. Grasp a Glass of Water Using NAO (Type: Rehabilitation)

Overview: This application [66] involves using a BCI-controlled humanoid to grasp a glass of water. This kind of application can be helpful for people who may find difficulty in performing such a task because of their age or a serious medical condition like Amyotrophic Lateral Sclerosis (ALS) disease. Note that ALS patients depend completely on caretakers for their daily needs. Scientists and researchers have always been actively looking forward to developing technologies to help such patients. A promising technology in this direction is the use of BCI-controlled humanoid robot. The authors in [66] use an EEG-based approach to capture the brain’s activity, which is recorded through electrodes implanted in cortical neurons. The signals were processed to actuate the humanoid to fetch the water. Salient state changes in their system are shown in Figure 5. The experiments for the BCI humanoid control for this task were performed by both healthy individuals as well as those suffering from ALS, and they was divided into multiple sessions, namely: (i) Calibration Session, (ii) Online Session, and (iii) Robotic Session. The purpose of dividing the experiment into multiple sessions was to tune the signal processing parameters as well as the classifier before performing the actual task in the Robotic Session. This is necessary because the parameters are dependent on the subject performing the tasks. This also helps the subjects to get familiar with the system. Description of each session is given in Table 3. Note that in Table 3, the threshold refers to the percentage of correct command selection that is required to transition from one session to the next one. Feedback indicates whether the visual feedback about the correctness of command was provided in the session. Accuracy is the ratio of correctly executed commands to the total number of commands. In this experiment an ERP approach known as the oddball paradigm [67] was used, which uses visual evoked potential. The oddball paradigm is an experimental design in which the subject is exposed to a sequence of repetitive stimuli which is infrequently interrupted by a deviant stimulus. The reaction of the subject to the oddball stimulus is recorded. In this case study, oddball paradigm is used to identify the infrequent visual stimuli that are elicited by highlighting the grid in the User Interface UI (Figure 6) of user’s interest. The P300 brain signals are eminent after approximately 300 ms of the stimulus.

System design: The system consisted of three major components. These were the user interface, the network interface, and the robotic system. The user interface used was a 3 × 3 matrix, as shown in Figure 6. Each grid in this figure represents an action performed by the humanoid. The interface shows two types of commands. The first set of commands are to control the movement of the humanoid robot in the environment, i.e., (forward, backward, turn, etc.) and the second set of commands are to grasp and give items. The grids showing the hand icon in Figure 6 correspond to the grasp and give actions, while the rest of the grids correspond to different movement commands. BCI data acquisition system, along with the user interface, collect the EEG signal using a g.USBamp EEG kit digitalised at 256 Hz. Various filters like notch and Butterworth filter were used to strengthen the signal and to remove the noise. The machine learning algorithm used for classification was stepwise LDA using the One vs. Rest approach. The One vs. Rest approach takes one class as positive and the rest as negative and trains the classifier. The One vs. Rest approach was used for selecting the class with the maximum distance from hyperplane compared to all the other classes [66]. The network interface passed the commands from the BCI system to the robotic system. The application part was completely dependent on the robotic system, which allowed two types of control modes. Both modes are illustrated in Figure 7.
Teleoperated Mode: In this mode, the user controls the movement of the robot and also gives commands to grasp and give a glass of water;Autonomous Mode: In this, the user would just give abstract commands and the humanoid plans its actions according to the state.

Results: The experiment showed that the BCI system, along with humanoid robots, can be effectively used by ALS patients with a mean accuracy of 71.25% in robotic session. Additionally, one of the interesting observation about the experiment reported by the authors was that the experimental setting (i.e., experiment conducted at home or with lab setting) did not affect the control performance significantly.

### 3.2. Telepresence by Humanoid Using P300 Signal (Type: Entertainment)

Overview: The application discussed in the previous section was simpler in terms of the actions performed, but provided a granularity of control that is sometimes not desired at the user level. This section discusses one such application in which a person is able to interact with the world using telepresence through a humanoid [68]. The control commands to be given to the humanoid in this case are high level, i.e., humanoid perform several subtasks that are grouped together and denoted as one high-level task (event, a few of such events can be seen in Figure 8a). Two major techniques used for the implementation of this application were (i) programming by demonstration in which the robot learns a task by observing someone performing it, and (ii) BCI-based control in which the brain signal generated by the visual stimuli is converted to control signals by classifying the P300 signal generated.

Overview: The application discussed in the previous section was simpler in terms of the actions performed, but provided a granularity of control that is sometimes not desired at the user level. This section discusses one such application in which a person is able to interact with the world using telepresence through a humanoid [68]. The control commands to be given to the humanoid in this case are high level, i.e., humanoid perform several subtasks that are grouped together and denoted as one high-level task (event, a few of such events can be seen in Figure 8a). Two major techniques used for the implementation of this application were (i) programming by demonstration in which the robot learns a task by observing someone performing it, and (ii) BCI-based control in which the brain signal generated by the visual stimuli is converted to control signals by classifying the P300 signal generated.

In this experiment, similar to the previous experiment (i.e., Section 3.1), the complete process was divided into two sessions illustrated in Table 4. The two sessions are namely: (i) Calibration session and (ii) real-time operation. The part of training the classifier was performed in the calibration session using the same EEG data, which in the previous case-study was performed in a separate session named online session. This experiment also used the oddball paradigm method for elicitation of the brain signals. However, as compared to the previous case study, the number of commands were increased to 16. All the commands used are high level, and are depicted in Figure 8a. The purpose of doing that was to remove the complexity of the humanoid control from the user end. Logistic regression was used for the classification of signals. It was used to train the function for predicting the output into the target or non-target events [68]. For the validation of the trained model, the subjects were asked to control the humanoids by brain signals. The set of tasks to be performed were pre-decided.

System Design: Figure 8b shows the abstract system design of the entire system. Some of the functionalities from the actual architecture have been grouped in the diagram to focus on key components. FieldTrip buffer is the main driver of the whole architecture, and it manages both, the BCI system as well as the NAO system. It also stores the BCI model. The subject uses the Graphical User Interface (GUI) to generate brain signals recorded using, g.USBamp, g.LADYbird with 256 Hz sampling frequency and 16 bit resolution. Signals are passed on to the BCI module for either tuning/training the model or for classification.

Results: During the calibration session, the model is trained and stored in the buffer. During real-time operation, the stored model is used to classify signals. Based on the classification, the events are generated and passed onto the NAO humanoid as control commands. The feedback of the same is shown on the user’s screen. The system achieved a real-time accuracy of 78% on average.

### 3.3. BCI Operated Museum Guide (Type: Entertainment)

Overview: This application [69] uses a remotely controlled robot that was operated by a healthy or paralysed person through BCI. The aim is to use the robot as a museum guide that will send remote visuals to the person operating it through BCI. In the application, the person could use the P300 signals to control the navigation of the robot. This provided the user with a perception of telepresence, similar to the previous case study. Note that although the authors did not use humanoid in their case study, a humanoid could very much be used in such an application, and thus the case study has been included. In this experiment, more focus was given on the GUI used in the BCI system. The GUI is different as it is more friendly for the user and is not aligned as a grid, like the UI used in previous case studies. The proposed BCI system used the P300 brain signal and the details about the BCI sessions are not discussed. In the new GUI, the selection of command was done by focusing on the flashing navigation arrow. This is similar to the oddball paradigm used in earlier experiments. To simplify the UI, the authors divided the process of selection into two parts. Each part has a different P300 elicitation interface. The first part is before starting with the input phase. In this, the user was asked to select between the two robots: Peoplebot and Pioneer3 depending upon the location they want to visit. In the application discussed, Peoplebot was located in the Computer Science department, and Pioneer3 was located in the Botanic garden. Both the robots were equipped with wheels for movement, micro-controller, IR sensors, sonar rings for avoiding collision and a camera. In general, the first part could be considered as a selection among two robots, Robot 1 and Robot 2, which were located at two different locations. The user could select the robot as per their preference to visit a location as shown in Figure 9a. After the selection of the robot, the navigational instruction was given using a screen, as shown in Figure 9b. The arrows represent the direction of the robot’s movement, which was continuous, and could be stopped using the stop button. All this was controlled using the brain signals based on P300. The screen in the middle displays the output generated using the robot’s camera.

System Design: The communication pattern between the robot and BCI System follows client-server architecture and Transmission Control Protocol/Internet Protocol (TCP/IP) is used in the network stack. Robot plays the role of the client, and the BCI system acts as a server. Initially, the robot tries to establish a connection with the BCI System and waits for the command to be executed. The BCI Architecture converts the signal from the brain into the corresponding command; the server then sends the command to the client program running at the robot end. The robot can handle three types of commands in general: (i) Start Session Command, (ii) Execution Command, and (iii) End Session Command. When the client-server connection is established “Start” command is received by the robot which enables direct control of robot through brain signals. This control is stopped by receiving the “End” command. At the server end, after sending the command to be executed, the server waits for the action to be executed. If the action is done, the server will get the result of the action from the client. However, if the command is not correct, the client will send a warning command to the server, and the server will respond by the same command.

Results: Using this application, a person could visit the museum through the robot because of telepresence. It was possible to simulate where the robot walked with the help of a two-dimensional map. The person could see the FOV (Field of View) of the robot’s camera with the help of a graphical user interface shown in Figure 9c and then decide the next displacement. Path planning could be done to avoid the sensor’s errors.

## 4. BCI-Controlled Humanoid Applications Using Hybrid BCI

In this section, in addition to the brain signals recorded using EEG, the control command is also dependent on complementary signals generated by some other parts of the body. We discuss two case studies in this section.

### 4.1. Picking Objects Using Neuro-Biological Feedback Fusion (Type: Rehabilitation)

Overview: The application [70] discussed in this section is similar to the one in which glass of water is fetched. However, the major difference is that this uses multi-sensor data for classifying the control commands. The authors discuss a new method for a human-humanoid interaction for ALS-affected patients. The authors make use of the biofeedback factor, which depends on the user’s intention, attention, and focus. This was then used to recognise the user’s mental state, based on which the robot was directed to do certain tasks.

The task performed in this application is very similar to [66]. Similarity can also be seen in the way the experiment was divided into Training Session, Online Session, and Robotic Session as discussed in Table 5. These sessions were combined with the biological feedback to support the decision making based on a certain threshold. The biological factors were used as it provides the mental state of the user. The architecture uses a combination of EEG signals which are elicited using visual stimuli along with a tracker that tracks the user’s eye movement. This biofeedback based system is used to extract features such as attention, intention and focus. Figure 10b shows the actual workflow. The task of the experiment was to grasp a glass of water.

System Design: NAO humanoid is used along with BCI system that includes a bio-signal amplifier which is used to convert the user’s brain signals into digital form and a tracker which tracks the location of the focus of user’s eye as shown in Figure 10a. Components of the System are as follows:BCI system:Visual Evoked Potentials (VEPs) and P300 are used. Oddball paradigm is used for eliciting ERPs. The salient features of the system were as follows:*Signal Processing:* g.USBamp device was used for recording the signals, using 10–20 standard system. The signal was digitised at 256 Hz. Butterworth filter was used to reduce the artefacts. A temporal filter was also used to average the samples in order to reduce the noise. In this study, 6 epochs each with a window of 800 ms were used.*Feature extraction:* Fisher’s stepwise Linear discriminant is used during the training to configure according to the user’s brain. LDA was used to differentiate the different classes by using hyperplanes. In this application, LDA calculates the stimuli recorded for every action on the grid and then selects the most prominent action corresponding to the grid.*User Interface:* It is similar to the 3 × 3 grid, which was used in [66] (Figure 6). Low-level behaviours include controlling all the possible directional movements of the humanoid. However, high-level behaviours include issuing control commands like holding some item and giving the held item, similar to the ones considered in [66].Biofeedback system uses neurological states and gaze: The biofeedback system takes into account the user’s eyes and brain activity. It includes four parameters—Mental intention, attention, visual focus, and stress. An action is executed only when the biofeedback factor (B${}_{f}$) is greater than 60%. The various modules associated with the bio-feedback system are explained below:*Attention module:* Since there are nine commands, Fisher’s Linear Discriminant (FLD) is used with *one versus rest* approach. The attention is expressed in percentage and is it based on the power of P300 waves measured during performing the task.*Intention module:* Correlation factor of the P300 wave is used to measure intention. It is based on the precision of the system.*Visual focus module:* It is calculated by evaluating the user’s gaze by eye-tracking, as shown in Figure 10a. Here F${}_{c}$ represents the central focus, F${}_{l}$ is the lateral focus, and F${}_{o}$ is the outer focus; all values are in the form of a percentage.*Entropy module:* Stressful Condition corresponds to high entropy in brain signals. Signal processing steps are performed to extract the normalised value of the entropy. Finally value B${}_{f}$ is calculated by taking a weighted average of attention, intention, and visual focus values.Connection of the subject to the robot: For receiving commands from the BCI, User Datagram Protocol (UDP) connection is made to the control interface. Connection to the robotic system is made through TCP/IP socket for reliability.***Controlling the behaviour of the robot.*** Two control modes are proposed by the authors: *Navigation mode:* NAO can move in 6 ways namely walking (front & reverse direction), turning ( left & right), and rotating (clockwise & anti-clockwise).*High-level mode:* It includes complex tasks like holding on to an object, and giving the object to the user after identifying the user’s location.The *distance metric* (O) is also used to avoid collisions based on a threshold value. If distance metric is less than the threshold value, then is considered safe to execute a command. Once that is ensured, corresponding to that an reaction safe command is activated along with the biological factor B${}_{f}$ and *O* which is passed to function which finally executes the command R${}_{k}$ that corresponds to the control command.

Results: In the experiment, the biological factor represents the mental state of the user. The average value of attention, visual focus and intention for healthy users during the online session were 74.59%, 99.03%, and 43.52%, whereas for ALS users the values were 76.70%, 90.81%, and 63.01%. During the robotic session the average values for these parameters for healthy users were 69.60%, 98.49%, and 42.98%, and ALS users achieved 79.45%, 96.16%, and 70.03% respectively. The attention and intention value for ALS users was better than healthy users. The B${}_{f}$ value also increased in the robotic session for ALS users. This denotes that the presence of robot in the robotic session acts as a positive feedback, particularly for ALS users, supporting studies like [71,72]. The same can also be attributed to better attention and intention among ALS users.

### 4.2. Humanoid Control using Facial Signals (Type: Entertainment)

Overview: This application [73] uses three types of bio-electric potentials, i.e., EOG (electric potential generated by eye movement), Glossokinetic Potential (GKP, the electric signals originated by tongue movement), and EMG. Although the application discussed here uses these three signals, as an EEG-based system is used for signal acquisition. Thus, the BCI data also can be made use of. With that integration, the system can utilise all the electric potentials generated from the entire head region. Application designed can identify two types of tongue movements, i.e., left-to-right and right-to-left, and two kinds of horizontal eye movements similar to the tongue movements, along with these two teeth-clenching movements generate EMG signals that are also used. By analysing these electric potential signals recorded from different parts of the face, a two-level interface is controlled. Eye movement selects a generic task category whereas the tongue movement selects a specific task from the category. Finally, teeth clenching executes the task. In the application, authors developed a mechanism that can detect and distinguish between the tongue and eye movements, and differentiate the direction of the movement of either tongue or eye. Basically, this means there are four types of movements which have to be distinguished accurately. These types are namely: (i) Tongue (left to right), (ii) tongue (right to left), (iii) eye (left to right), and (iv) eye (right to left).

System Design: The experiment consisted of two phases, training and online. Table 6 consist of more details. For the training part, both eye and tongue movement were recorded for seven rounds (trials). g.Mobilab device was used for recording. This device has the facility of recording EEG, EOG, EMG, and GKP singals in this experiment. The signals were digitised at 256 Hz and filtered above 0.5 Hz using the high-pass filter.

For eye-movement, auditory cues were used to guide the user, whereas visual cues were used in case of tongue movement. A RBF-SVM (Radial Basis Function-SVM) model was trained for classifying the four kinds of movement. It was used because it has an enclosed decision boundary and can be used to reject irrelevant artefacts generated due to the motion of the electrodes. The distinction between tongue and eye movements was obtained using PCA based feature extraction. For the online part, the authors evaluated the experiments in terms of: (i) Performance (accuracy and response time), (ii) task execution (this method has been extensively used in other case studies as well for evaluation, in which the user is asked to perform a set of tasks on the robot), and (iii) workload (to measure qualitative parameters). Figure 11a shows the two-level hierarchical menu displayed on the user screen to allow them to control the interface, as shown in Figure 11b. All the similar tasks are grouped in the two-level interface under a category. By default, the task in the category at the central position of the screen is highlighted which can be executed by a teeth-clenching movement resulting in the generation of EMG signal. For navigation among the categories, eye movements (left to right) and vice versa are used. Furthermore, for navigation within a category, tongue movements (left to right) and (right to left) are used. Eye: Left to right movements moves the category selection in the clockwise direction, whereas the right to left movement will move in an anticlockwise direction. Within task categories, a specific task was selected by the tongue movements. After the selection of the task was made, the execution was done by teeth clenching movement. All the categories and one of the task used in [73] along with the transitions are shown in Figure 11b.

Results: The mean accuracy of the system was 86.7 ± 8.28% with an average response time of 2.77 ± 0.72 s. This scheme can be supported with facial recognition for expression recognition [74] and can be integrated with some of the action commands to increase robustness.

## 5. Application Using BCI Supported by Augmented Reality (AR)/Virtual Reality (VR)

In this section, the application discussed uses augmented reality to create a sense of embodiment and is used to have greater control over the environment.

### Navigational Assistance using AR & BCI (Type: Rehabilitation)

Overview: In this application discussed in [75], a novel navigation scheme is presented to control a humanoid through BCI enabling it to interact with the environment. SSVEP signals are used in this study. For interaction with humans, a high level of accuracy is desired. This is achieved using a sequence of manual and automated phases presented in the assistive navigation scheme. HRP-2 robot is used in this demonstration.

The authors focus majorly on demonstrating a new navigation scheme that is assisted with a Head-Mounted Display (HMD) to increase the sense of embodiment by displaying the robot’s camera video feed to the user. The humanoid control is done by generating control commands using the SSVEP paradigm. The elicitation of SSVEP is also done with the help of HMD. The navigational assistance is achieved by executing a sequence of manual and automated phases. In general, the selection-based phases are assigned to the user, whereas navigation and interaction-based tasks are automated to achieve high-level accuracy while interacting with humans.

System Design: The experiment [75] is divided into five phases, as shown in Figure 12.

Major characteristics of these phases are listed below:*Manual navigation phase*—This is a manual phase that requires the task to be performed by a user. The phase is limited to the user locating himself using the robot’s camera. The output of the camera is visible in the HMD;*Body part selection phase*—This phase is also performed by the user manually. In this phase, the user selects the body part which the humanoid robot is expected to interact with;*Assistive navigation phase*—This is an automated phase. The Robot uses SLAM [76] to navigate towards the selected body part. The experiment also shows that this kind of navigation is better because of the difficulty associated with manual navigation which causes errors in navigation along with slow execution of the task;*Interaction selection phase*—This is a manual task. The user selects the type of interaction on the body part selected;*Interaction phase*—This is an automated phase. The humanoid performs minor adjustments to perform the interaction. In this particular application, a user’s arm is touched. But in general, any task can be configured in the humanoid, and it will execute the task when triggered.

The navigational assistance system consists of a HMD which is responsible for displaying live video feed and for the elicitation of SSVEP signals to generate control commands. AR markers were placed on the HMD and user arms which helps in performing the automated phases. As shown in Figure 13a, SSVEP was evoked by flickering the body parts, which was used for body part selection by the user. g.USBamp was used to acquire the data with a sampling rate of 256 Hz combined with band-pass filter (0.5–30 Hz) and notch filter (50 Hz). Similarly, SSVEP was evoked during the interaction selection phase as well. Finally, as shown in Figure 13b the robot adjusted itself by small steps. The robot initiated the action when it reached a comfortable pose.

Results: The task for this application was touching the user’s arm, as shown. The system operated at an accuracy of more than 80% with a training of about 6 min.

## 6. Summary of Applications

In this paper, we discussed various applications that deal with controlling a humanoid with the help of BCI signals. These experiments were performed using various humanoids and different translation algorithms were used to generate the control signals. Table 7 presents the summary of the studies considered in this review.

## 7. Conclusions

BCI has emerged as a new communication system and is an active field of research. This paper discussed BCI-controlled humanoid applications of three kinds: a. The ones using just EEG signals, b. using Hybrid BCI, and c. Augmented reality-assisted BCI humanoid control. Section 3 discussed three applications that make use of P300 signals as an input for classification. These signals were generated using a grid like user interface denoting different actions. Section 4 covered two application which combine input from multiple sensors to increase the robustness of the system. The application performed in Section 3.1 and Section 4.1 are similar. However, the application in Section 4.1 used neuro-biological feedback to accomplish the task, and had better accuracy on account of using multiple inputs. Application in Section 5 used augmented reality to demonstrate a navigation scheme that could be controlled from a head mounted display. Most of the applications discussed in this paper deals with increasing the quality of life of a person with paralysis or motor impairment, though it could also be beneficial for a healthy person in some cases. Current applications have experimented with objectives ranging from, accompanying a patient to fetch a glass of water using humanoids to using augmented reality for humanoid control. Major issues faced while implementing each of the applications was the process of training and calibration which takes time. Most of the complimentary techniques deal with reducing the training time and improving the online accuracy while performing the action. This paper reinforced the fact that BCI could be used to control the humanoid with a good amount of accuracy. In most of the applications discussed, this was achieved by dividing the experiment into phases and having an initial training phase to tune the model according to the subject.

## Figures and Tables

**Figure 1 sensors-20-03620-f001:**
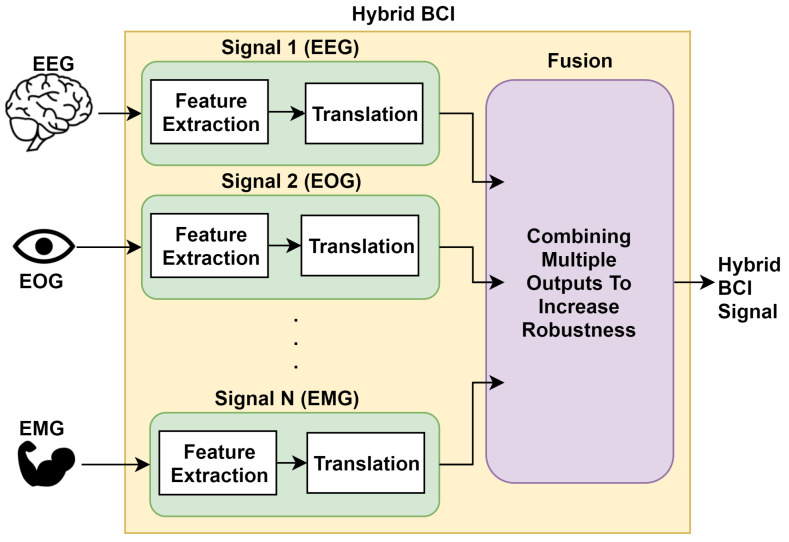
Block diagram of Hybrid Brain-Computer Interface (BCI).

**Figure 2 sensors-20-03620-f002:**
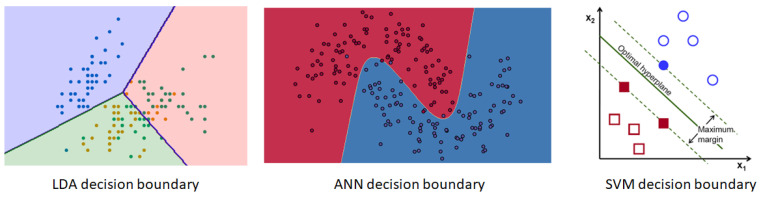
Decision boundaries for the different classifiers (Linear Discriminant analysis (LDA), Support Vector Machines (SVM), and Artificial Neural Networks (ANN)).

**Figure 3 sensors-20-03620-f003:**
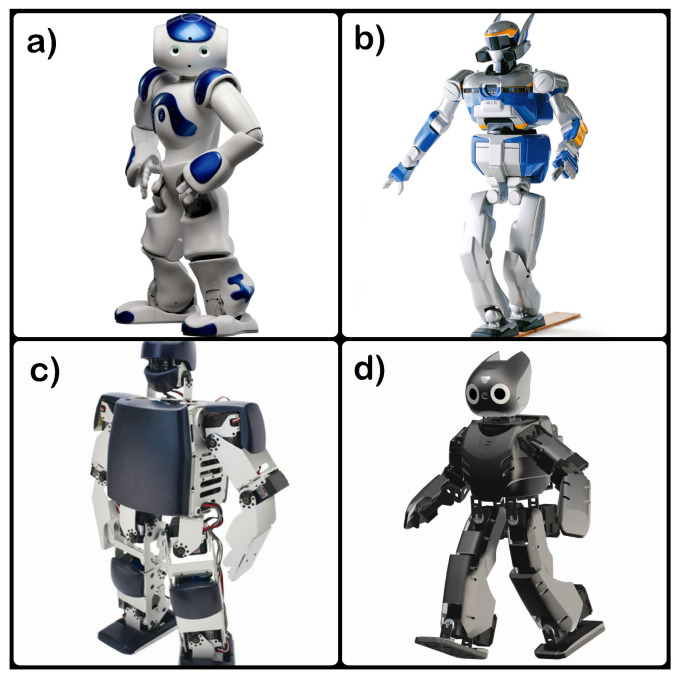
Humanoids: (**a**) NAO (Nao Humanoid), (**b**) HRP 2, (**c**) KT-X, and (**d**) DARwIn-OP.

**Figure 4 sensors-20-03620-f004:**
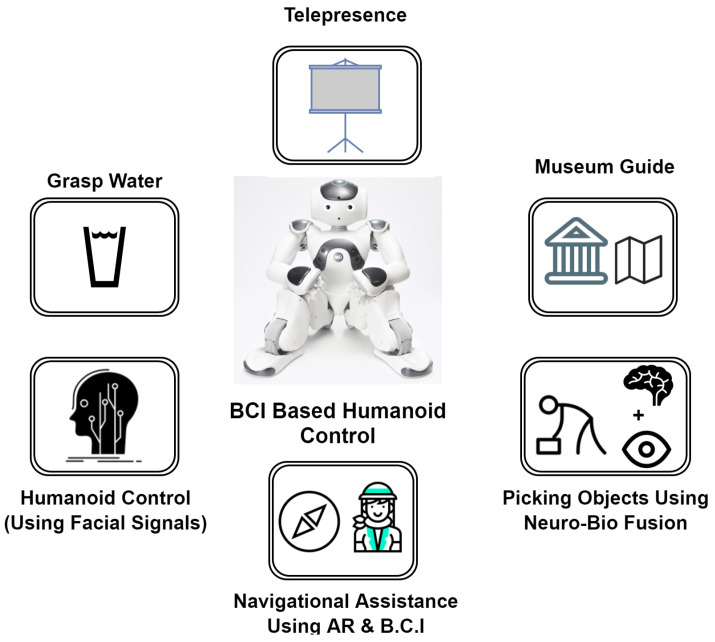
Overview of applications.

**Figure 5 sensors-20-03620-f005:**
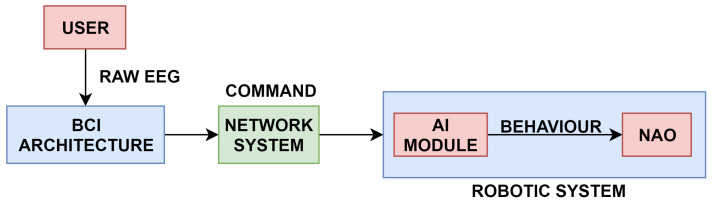
State diagram of process (adapted from: [66]).

**Figure 6 sensors-20-03620-f006:**
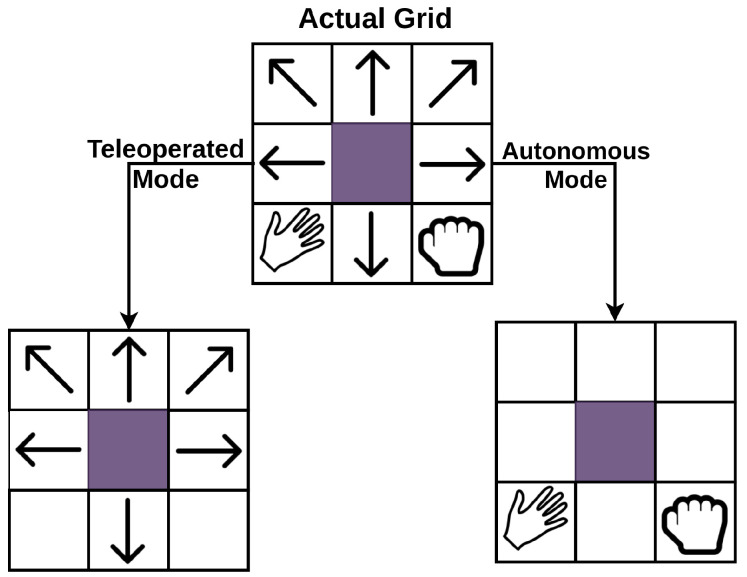
3 × 3 matrix showing user interface (adapted from: [66]). (**left**) Teleoperated Mode: User gives directional command using only arrows and (**right**) Autonomous Mode: User gives high-level commands corresponding to the symbol.

**Figure 7 sensors-20-03620-f007:**
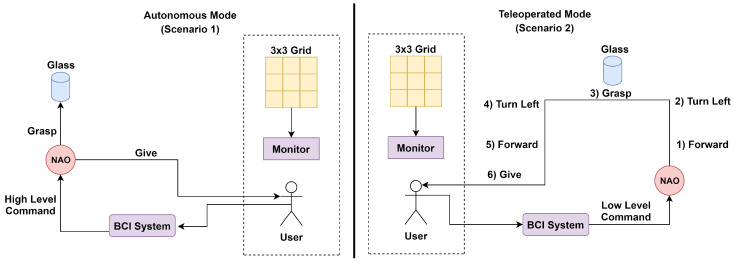
Autonomous and teleoperated mode (adapted from: [66]).

**Figure 8 sensors-20-03620-f008:**
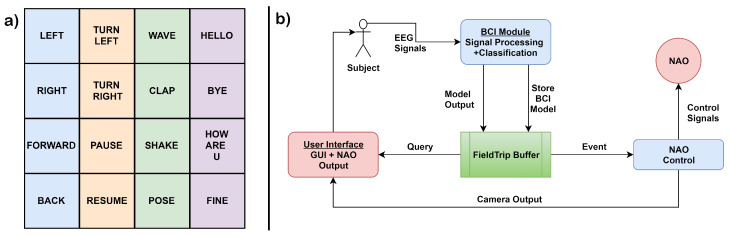
(**a**) 4 × 4 Grid showing high-level commands and (**b**) abstract system pipeline for telepresence (adapted from: [68]).

**Figure 9 sensors-20-03620-f009:**
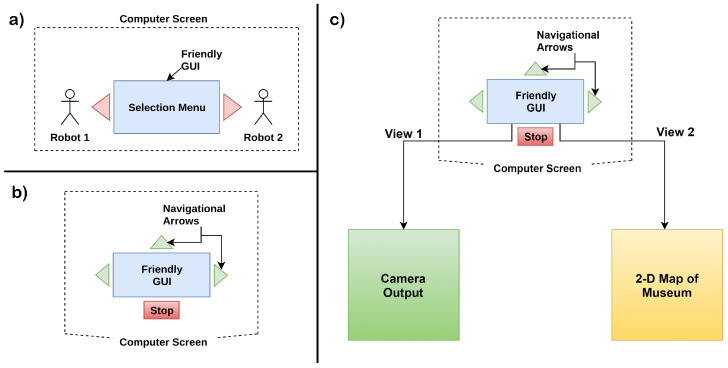
(**a**) Robot selection menu, (**b**) navigation screen, and (**c**) two views for the user (adapted from: [69]).

**Figure 10 sensors-20-03620-f010:**
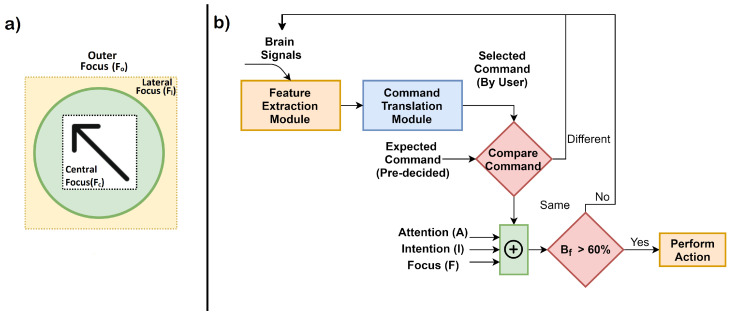
(**a**) Eyeball tracking in grid cell, (**b**) flow chart of the system using neuro-biological fusion (adapted from: [70]).

**Figure 11 sensors-20-03620-f011:**
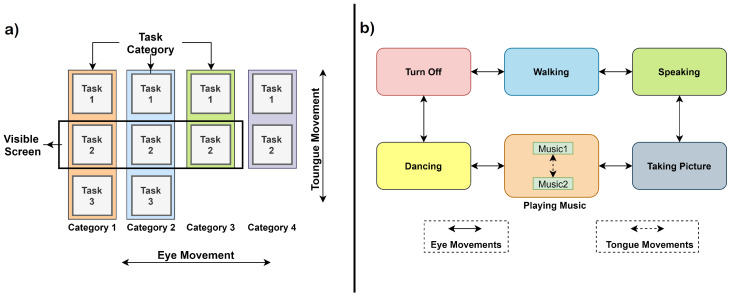
(**a**) Menu for selecting task, and (**b**) state diagram (adopted from: [73]).

**Figure 12 sensors-20-03620-f012:**
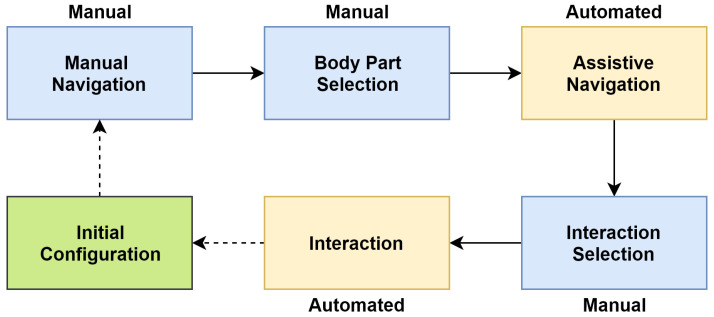
State diagram of assistive navigation (adapted from: [75]).

**Figure 13 sensors-20-03620-f013:**
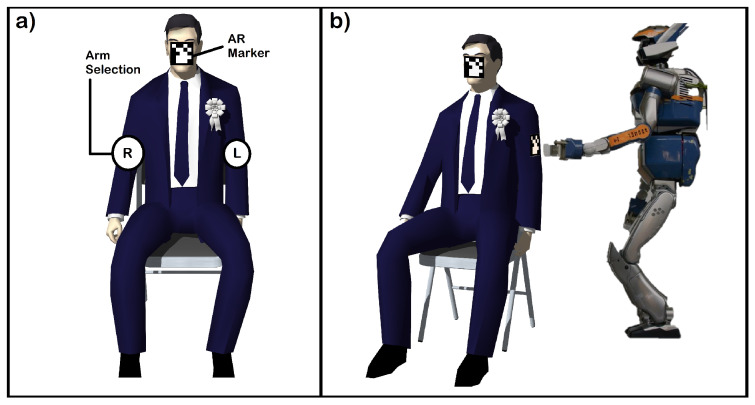
(**a**) SSVEP for arm selection and (**b**) interaction phase (selected arm is touched).

**Table 1 sensors-20-03620-t001:** Comparative Analysis of Various Methods used for Recording Features.

S.No.	Method		Description	Characteristics
(1)	Electroencephalography (EEG)		Measuring the electric signals produced by the human brain	- Commonly used method. - Safe and affordable - Poor spatial resolution
	(a) Evoked Signals	SSVEP	Brain signal generated in response to looking at source having a specific frequency of flickering	- Training time is short - Requires continuous attention for stimuli - Exhausting for user after long sessions
		P300	Signal generated in response to an infrequent stimulus, recorded with a latency of 250–500 ms	
	(b) Spontaneous Signals		Voluntary signals generated without external stimulus	- External stimuli not required - Long training required
(2)	Electromyography EMG)		Measure the electrical activity produced by skeletal muscles	- Easy to record - More noise contamination
(3)	Electrocorticography (ECoG)		Measuring the electric signals by placing electrodes beneath the skull	- Better signal quality than EEG - Risky (semi-invasive) - Less Common
(4)	Functional magnetic resonance imaging (fMRI)		Measure changes in the metabolism of the brain (e.g., oxygen saturation)	- Good spatial resolution - Poor temporal resolution(1 s–2 s) - Sensitive to motion
(5)	Near-Infrared Spectroscopy (NIRS)			- Good spatial resolution - Poor temporal resolution(2 s–5 s)

**Table 2 sensors-20-03620-t002:** Comparison of classification algorithms.

Classifier	Mechanism	Properties	Choice Consideration
Linear Discriminant Analysis (LDA)	Decision boundary is made by maximising the mean among two class and minimising the variance inside each class.	1) Simple 2) Less computational 3) Decision boundary is linear	- Suited for online sessions - Smaller training set
Artificial Neural Networks (ANN)	Minimises the error in classifying training data by adjusting weights of neural connections	1) Many parameters to set 2) Highly computational 3) Decision boundary is non-linear 4) Prone to overfitting	- Suitable for variety of applications - Sensitive to noisy data
Support Vector Machines (SVM)	Decision boundary maximises the margin between two class	1) Decision boundary can be linear or non-linear 2) Less prone to overfitting 3) High computation for non-linear cases	- Appropriate for high- dimensional data - Less sensitive to noisy data
Statistical Classifiers	Estimates probability corresponding to each class and selects the class having the most favourable possibility	1) Decision boundary is non-linear 2) Efficient for uncertain samples.	- Suited as adaptive algorithm - Considers variation in brain dynamics (e.g., fatigue)

**Table 3 sensors-20-03620-t003:** BCI Sessions used in [66].

Session	Trials	Threshold & Feedback	Purpose	Accuracy (In %) (Mean ± Standard Deviation
Calibration	9	100% No Feedback	For tuning signal processing parameters	-
Online	20	55% With Feedback	Train the classifier	Healthy: 74.5 ± 5.3 Amyotrophic Lateral Sclerosis (ALS) Patient: 69.75 ± 15.8
Robotic	10	N.A. With Feedback	Robot Executes the selected command	Healthy: 72.4 ± 9.4 ALS Patient: 71.25 ± 17.3

**Table 4 sensors-20-03620-t004:** Experiment sessions used in [68].

Session	Trials	Feedback	Purpose	Accuracy (In %)
Calibration	5	With Feedback	Tune Signal Processing Parameters & Train Classifier	-
Real-Time	-	With Feedback	Control the Humanoid Robot.	78

**Table 5 sensors-20-03620-t005:** Experiment sessions used in [70].

Session	Trials	Threshold & Feedback	Purpose	Success	Bio-Feedback Factor
Calibration	Till 100% correctness (Avg.: 3)	100% No Feedback	Calibrate BCI System over the neural response	-	-
Online	10	- With Feedback	Select the command with visual feedback	Healthy: 100% ALS: 97.22%	Healthy: 78.15% ALS: 79.61
Robotic	5	- With Feedback	Select the command with robotic feedback	Healthy: 100% ALS: 96.97%	Healthy: 75.83 ALS: 84.25

**Table 6 sensors-20-03620-t006:** Phases of experiment in [73].

Session	Trials	Purpose	Accuracy
Training	7 (Eye & Tongue)	To train the detection model	-
Online	1	To evaluate the performance of the system	86.7 ± 8.28%

**Table 7 sensors-20-03620-t007:** Summary of applications.

Name	Related Works	Used Signal	Classifier	Humanoid Used	Description
Fetching Water (Rossella et al., 2017) [66]	[77,78,79,80]	P300	Stepwise LDA	NAO Humanoid	Humanoid fetches a glass of water for a patient using BCI-P300
Telepresence (Batyrkhan et al., 2018) [68]	[81,82,83,84,85,86,87]	P300	Logistic Regression	NAO Humanoid	A user can interact with the world remotely using humanoid controlled by BCI
Museum Guide (Antonio et al., 2009) [69]	[88,89]	P300	N.A.	PeopleBot & Pioneer3	A user can control a robot to visit a museum remotely
Picking Object (Bio-Feedback) Rosario et al., 2018) [70]	[90,91,92,93]	P300 + Eyeball Tracking	Stepwise LDA	NAO Humanoid	Picking & placing objects. But control signals are generated based on bio-logical feedback & brain signal
Control by Facial Signal (Yunjun et al., 2014 [73]	[94,95,96,97,98,99]	EOG, EMG, GKP	SVM	NAO Humanoid	Humanoid is controlled by facial signals which do not depend on spine for signal delivery
Navigational Assistance (Damien et al., 2014) [75]	[100,101,102,103,104,105,106]	SSVEP	N.A.	HRP-2 Humanoid	A navigational scheme is presented to have greater precision while performing action using humanoid

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
