# Peer review of "Brain-Computer Interface-Based Humanoid Control: A Review"

_sensors, 2020, doi:10.3390/s20133620_

Round 1
Reviewer 1 Report
This review of the BCI methods is interesting and generally well-written. I have some conceptual points to address.
In general, there are several typos. Just as an example, line 42 "organised", while in the rest of the manuscript they choose the American spelling, like "recognized". Again line 42, and in the whole section, consider the plural for the word "application". Line 45 "discuss" should be "discusses". And so on. Consider the use of hyphen for composed word like, for example, interaction-based or selection-based (lines 392 and 393). Sometimes a missing comma is required, especially at the beginning of the sentence, after, for example, "In this paper," (line 419).
Please, revise the language of the entire manuscript.
Introduction:
In the first two sentenced the BCI is introduced as an "emerging field", then as a method.
First, I think those two concepts are in contradiction since a field gathers more techniques and methodology. Second, I don't think that "emerging" can be a valid description, since maybe the first control of a system using the brain waves is probably due to William Walter Grey (who died in 1977).
Section 2
Can the authors please explain the sentence in line 52? I think this consideration of independency can confuse the readers.
Section 2.3 (humanoids) appears to be out of place. You are describing theoretically concepts, I suggest to move this part at the beginning of section 3, maybe as an introduction
Section 4
At some point, authors introduced several equations, most of which are pretty useless, since there is not a real mathematical theoretical approach to the theory explained, and most of them appear out of the blue. I suggest trying to reduce the number, in order to improve the readability of the paper.
Section 6
Table 7 is pretty hard to read, because of the references. Please consider using the classic citation, "First name et al., year" at least for this table.
Reviewer 2 Report
The authors explore a series of Hybrid BCI systems on humanoid control. I think this is an interesting topic. But the main concern is that many mentions references are just conference papers. There are doubts about research rigor and manuscript quality. Other comments are listed as follows.
- line 80: “To maximize the robustness of the system, increase the information transfer rate and decrease the training time current BCI system are driven by recording and analysing various complementary signals”. But the performance (robustness, ITR, training time, etc.) of the mentioned Hybrid BCI systems were not presented in the manuscript. The authors should describe the performances of the mentioned hybrid BCI system in detail. Could the authors also compare with the performance of the BCI system with single input of brain signal?
- Table 1 mentions several brain control signals. In fact, SSVEP, ERP, P300,… are just a kind of EEG signals. The authors should re-summarize Table 1. Moreover, the characteristics of these brain signal, such as advantage, disadvantage, limitation,… , should also be presented.
- In section 2.4, most of classifiers are well-known. The authors should simplify the descriptions of these classifiers. But the main characteristics of these classifiers, such as advantage, disadvantage, limitation,…, etc., were not presented in the manuscript. The authors should describe these classifier characteristics and the choice considerations of classifier types, and re-summarize these information in Table 2.
- Line 163: “Majorly P300 signal is used in these applications as it gives high accuracy.” This description has been carefully confirmed. The year of the cited reference is 2009, and it is not new enough.
- In sections 3 and 4, the content structures for the introduction of each hybrid BCI systems on humanoid control must be consistent. In additions, the authors should add the specific hardware specifications, algorithms, performances and other evaluation statements.
- The comparison of the hybrid BCI system is not detailed enough. There should be a clear comparison of performance or advantages and disadvantages, … etc.
Reviewer 3 Report
This review paper aims at summarizing the main advances in brain-computer interfaces for controlling humanoid robots. The topic is interesting, and the authors have included and analyzed a number of studies. Yet, I think the manuscript requires significant improvement to make it suitable for publication.
The major concern I have is about organization of the paper and language. A review paper lacks in novel research and, as such, it should strengthen the way it presents concepts. The current form of the manuscript does not read well, and it looks more like a set of subsections put together, each subsection briefly summarizing each analyzed paper. It lacks flow and links between sections, it contains a lot of repetitions that make it unnecessarily long.
The contribution of the paper should also be made clearer. Why do authors focus specifically to humanoid robots, and not to robots in general?
The authors should broaden their understanding of BCIs and revise the introduction. BCIs are not an emerging field, they have been around since the 70s. A BCI is not a method, it is a device, and is not only based on EEG, it could be based on other neural recordings too (for example, fNIRS, ECoG...). Several concepts described in the preliminary knowledge are wrong. For example, in Table 1, the P300 is not the person's response to a stimulus, that is an event-related potential. The P300 is a particular type of ERP that is elicited with specific rare stimuli. In Table 2, the mechanism of ANN "Minimizes the error in classifying training data" is too generic and valid for most supervised machine learning approaches. Traditional BCIs do not rely on "a single feature", they rely to a single type of brain signal (e.g., EEG).
The authors should consider proofreading the manuscript with a native-English speaker, as it is difficult to read in the current form. For example, the determiner "The" is missing before several nouns (e.g., the paper, the rest of the the paper, ...). "Complimentary" means free, while "complementary" is the word you want.
Figure 1 has been copied with minor changes from Thakor 2013 without proper citation. This is plagiarism, so the authors must either remove the figure or ask Dr. Thakor and Science Translational Medicine for authorization.
The paper has too many figures and tables, and most of them are not very relevant (e.g., Fig 2, 3, Fig 22 and 23 could be combined, Table 4, 6). Many figures have been taken by other manuscripts, without referring to them in the caption (again, plagiarism).
Round 2
Reviewer 1 Report
The Authors answered all my comments and therefore I recommend publication
Author Response
Thank you for taking out time to carefully review our paper and helping us improve the quality of the paper in the previous round.
Reviewer 2 Report
Although many mentions references are still conference papers, the authors have carefully modified the manuscript structure, and this topic is interesting. I suggest it may be considered for sensors publication after English modification.
Author Response
Thank you for taking out time to carefully review our paper and for your kind comments. We have got the manuscript again proof-read and necessary corrections have been made to the best of our knowledge.
Reviewer 3 Report
I thank the authors to have made an effort in addressing my comments.
I could not find where the authors have indicated the contributions of this manuscript. A separate section or paragraph in the introduction will help, possibly using the word "contribution".
I believe the manuscript still lacks flow and is not a well organized review. For example, grouping the applications by type (currently listed in parenthesis in each subsection) may streamline the manuscript and help with the flow. The authors should comment on each application and summarize what is interesting. Also, they should extend the conclusion section to provide a further summary of the analyzed literature.
Regarding the figures, the manuscript still has many figures copied from other manuscripts, and the authors should get permission to republish them from publishers.
The manuscript would still benefit from a professional proofreader, as it contains mistakes that do not help with the flow. For example, even the first sentence "Brain-Computer Interface (BCI) lies..." should either be "A Brain-Computer Interface (BCI) lies" or "Brain-Computer Interfaces (BCIs) lie".
Author Response
Please see the reply in attachment
